# Enhancing the Photocatalytic Activity of Halide Perovskite Cesium Bismuth Bromide/Hydrogen Titanate Heterostructures for Benzyl Alcohol Oxidation

**DOI:** 10.3390/nano14090752

**Published:** 2024-04-25

**Authors:** Huzaikha Awang, Abdo Hezam, Tim Peppel, Jennifer Strunk

**Affiliations:** 1Leibniz Institute for Catalysis, Albert-Einstein-Str. 29a, 18059 Rostock, Germany; huzaikha.awang@catalysis.de; 2Preparatory Centre for Science and Technology, Universiti Malaysia Sabah, Jalan UMS, Kota Kinabalu 88400, Sabah, Malaysia; 3School of Natural Sciences, Technical University of Munich (TUM), Lichtenbergstr. 4, 85748 Garching, Germany; abdo.mohsen@tum.de

**Keywords:** benzyl alcohol oxidation, halide perovskite, hydrogen titanate, nanosheet, photocatalysis

## Abstract

Halide perovskite Cs_3_Bi_2_Br_9_ (CBB) has excellent potential in photocatalysis due to its promising light-harvesting properties. However, its photocatalytic performance might be limited due to the unfavorable charge carrier migration and water-induced properties, which limit the stability and photocatalytic performance. Therefore, we address this constraint in this work by synthesizing a stable halide perovskite heterojunction by introducing hydrogen titanate nanosheets (H_2_Ti_3_O_7_-NS, HTiO-NS). Optimizing the weight % (wt%) of CBB enables synthesizing the optimal CBB/HTiO-NS, CBHTNS heterostructure. The detailed morphology and structure characterization proved that the cubic shape of CBB is anchored on the HTiO-NS surface. The 30 wt% CBB/HTiO-NS-30 (CBHTNS-30) heterojunction showed the highest BnOH photooxidation performance with 98% conversion and 75% benzoic acid (BzA) selectivity at 2 h under blue light irradiation. Detailed optical and photoelectrochemical characterization showed that the incorporating CBB and HTiO-NS widened the range of the visible-light response and improved the ability to separate the photo-induced charge carriers. The presence of HTiO-NS has increased the oxidative properties, possibly by charge separation in the heterojunction, which facilitated the generation of superoxide and hydroxyl radicals. A possible reaction pathway for the photocatalytic oxidation of BnOH to BzH and BzA was also suggested. Furthermore, through scavenger experiments, we found that the photogenerated h^+^, e^−^ and •O_2_^−^ play an essential role in the BnOH photooxidation, while the •OH have a minor effect on the reaction. This work may provide a strategy for using HTiO-NS-based photocatalyst to enhance the charge carrier migration and photocatalytic performance of CBB.

## 1. Introduction

Photocatalytic technology has induced global interest in offering a green path for simultaneously addressing global issues, such as future energy crises and unavoidable CO_2_ emissions [1]. Photocatalytic technology has been applied in various areas, such as H_2_ evolution, CO_2_ reduction, N_2_ fixation, photocatalytic synthesis and environmental remediation [1,2]. Photocatalytic synthesis, such as alcohol oxidation, aldehyde alkylation, thiol reaction and dehydrogenation, are essential in our modern life to produce carbonyl compounds such as food additives, fragrances and many value-added organic chemicals [3,4]. The formation of carbonyl compounds using photocatalytic technology requires the utilization of photocatalysts without being consumed. Due to the photoelectric effect, the photocatalyst, also known as a semiconductor, can adsorb photons and generate the electron–hole pair (exciton) to initiate the reaction. Various semiconductor materials have been explored for photocatalytic synthesis in the past decades. However, the low light-harvesting capability and photoconversion efficiency retain such processes in infancy [5,6,7,8,9]. Perovskite materials have emerged as one of the most promising candidates owing to their excellent optical absorption, adjustable band gap and low-temperature requirement [10,11].

An organic–inorganic hybrid bismuth-based perovskite MA_3_Bi_2_X_9_ (MA = CH_3_NH_3_^+^, Cs^+^; X = Cl, Br, I) has been successfully explored and developed [12,13,14]. It has also been reported that the organic halide perovskite such as cesium bismuth halide (Cs_3_Bi_2_X_9_, X = Cl, Br, I) is more beneficial due to its low cost, low toxicity and good stability against air, light and heat compared to the other inorganic halide perovskites [15]. Since Park et al. first reported the photocatalytic hydrogen evolution of MAPbI_3_ [16], the application of perovskite materials employed as photocatalysts has developed remarkably in various applications such as pollutant photodegradation [17], CO_2_ photoreduction [18], including selective benzyl alcohol photooxidation [10]. However, the practical application of halide perovskite is still hindered due to its instability in a polar solvent (such as water) and low charge carrier migration [17]. The low charge carrier migration and short lifetime of halide perovskite should be addressed because it led to recombining the photogenerated charges rather than participating in the photocatalytic reaction [19,20,21]. The heterojunction construction is a feasible method to promote carrier separation efficiency [21,22,23,24].

Designing a suitable heterojunction structure can improve light harvesting and absorption abilities and, therefore, can enhance the redox abilities of the catalyst, too [22,23]. Suitably matching band structures between different materials can form internal electric fields, which help to promote the transfer of photogenerated charge carriers and facilitate charge separation. Recently, hydrogen titanate nanomaterials (HTNMs) in the form of nanosheets (HTNS) with the chemical notation of H_2_Ti_x_O_2x+1_ have received significant attention in various applications due to their unique morphology, physicochemical properties, high specific surface area, adjustable band structures and the ability to exchange cations in aqueous solution [24,25,26]. The excellent properties of HTNMs made them a good candidate for the photocatalyst base to construct the heterostructure in this study. Stefan S. et al. reported that CsPbBr_3_/TiO_2_ showed enhanced benzyl alcohol oxidation compared to pristine perovskite [27]. However, the toxic effects of lead-based halide perovskites might endanger the ecosystem. Qimeng Sun et al. reported in 2021 that the heterojunction Cs_3_Bi_2_Br_9_/TiO_2_ shows an enhanced photocatalytic efficiency of benzyl alcohol oxidation [28]. Wei-Long et al. reported in 2024 that the composite of C_3_N_4_/Cs_3_Bi_2_Br_9_ significantly improved the photocatalytic degradation efficiency of RhB by 98% at 1 h [29]. Despite the development of halide perovskite heterostructure, the low charge carrier migration and photocatalytic performance are still of continuous concern.

In this research article, we present a straightforward approach to address the low charge carrier migration and photocatalytic performance of CBB by synthesizing a stable Cs_3_Bi_2_Br_9_/HTiO-NS (CBHTNS) heterostructure. By introducing HTiO-NS via a modified anti-solvent reprecipitation method, we have successfully synthesized the optimal CBHTNS heterostructures by optimizing the amount of CBB. By widening the range of visible-light response, improving the charge carrier migration and enhancing the oxidative properties, we aim to improve the photocatalytic performance of CBHTNS. We successfully applied the heterojunction in the selective photocatalytic oxidation of benzyl alcohol under blue light, which is an exemplary case of our research. We observed that the 30 wt% Cs_3_Bi_2_Br_9_/HTiO-NS (CBHTNS-30) heterojunction showed the highest benzyl alcohol photooxidation performance with 98% BnOH conversion and 75% benzoic acid (BzA) selectivity. Therefore, through the successful synthesis and detailed characterization, we shed light on the important role played by the CBHTNS heterostructure. The hydrogen titanate material (HTNM)-based photocatalyst has yet to be introduced to the halide perovskite to construct a stable heterostructure. This work may provide a strategy for using HTNM-based photocatalysts to enhance CBB’s charge carrier migration and photocatalytic performance. Furthermore, through scavenger experiments, we found that the photogenerated h+, e^−^ and •O_2_^−^ play an essential role in the BnOH photooxidation, while the •OH have a minor effect on the reaction.

## 2. Materials and Methods

### 2.1. Materials

Hombikat UV100 (UV100, Venator, Wynyard, UK), sodium hydroxide (NaOH, Fisher Scientific, Hampton, NH, USA), hydrochloric acid (HCl, 37%, Fisher Scientific) and deionized water (DI) were used to synthesize pristine hydrogen titanate. Bismuth bromide (BiBr_3_, Sigma-Aldrich, St. Louis, MO, USA), cesium bromide (CsBr, Sigma-Aldrich), dimethyl sulfoxide (DMSO, Sigma-Aldrich) and isopropanol (IPA, Sigma-Aldrich) was used to synthesize pristine Cs_3_Bi_2_Br_9_ and Cs_3_Bi_2_Br_9_/Hydrogen Titanate heterostructures. Benzyl alcohol (BnOH, Sigma-Aldrich), Acetonitrile (MeCN, Sigma-Aldrich) and 1,2-dichlorobenzene (DCB, Sigma-Aldrich) were used for benzyl alcohol oxidation experiments. Potassium iodide (KI, Sigma-Aldrich), silver nitrate (AgNO_3_, Sigma-Aldrich), *p*-benzoquinone (p-BQ, Sigma-Aldrich) and tert-butyl alcohol (TBA, Sigma-Aldrich) were used for scavenger tests to identify the main reactive oxygen species (ROSs) responsible for BnOH conversion. All chemical reagents were used in purities >99% as received without any further purification.

### 2.2. Synthesis of HTiO-NS

The preparation of hydrogen titanate nanosheets (HTiO-NS) is illustrated in Figure 1a as reported in the literature with minor modifications [30]. First, 1.00 g of UV100 powder was added into 70 mL 10 M NaOH solutions and ultrasonicated at room temperature for 40 min. After the sonication, the mixture was further stirred at room temperature for 1 h at 500 rpm. The mixture was then transferred into an autoclave which was closed tightly and held for 24 h at 110 °C. After hydrothermal reaction, the freshly obtained hydrogen titanate was washed thoroughly with portions of DI water and 0.1 M HCl aqueous solution. The washing procedure was repeated until the filtrate showed pH < 7. The obtained white solids were oven-dried at 70 °C and stored as a HTiO-NS. The mass of the synthesized HTiO-NS was 0.81 g. 

### 2.3. Synthesis of Cs_3_Bi_2_Br_9_ (CBB)

The preparation of Cs_3_Bi_2_Br_9_ (CBB) is illustrated in Figure 1b without adding the HTiO-NS. CBB was prepared via a modified antisolvent reprecipitation method at ambient conditions using CsBr and BiBr_3_ as previously reported in the literature with minor modifications [31]. First, 192.0 mg of CsBr (0.9 mmol) and 270.0 mg of BiBr_3_ (0.6 mmol) were dissolved in 20 mL DMSO. The CBB precursor solution was then stirred at 500 rpm until there were no remaining solids. Then, the precursor solution was injected quickly into 500 mL IPA under vigorous stirring at room temperature and was further continuously stirred for 10 min. Next, the obtained mixture was centrifuged at 3000 rpm for 3 min to precipitate the larger particles and then centrifuged at 10,000 rpm for another 5 min using IPA. The obtained yellow solids were then vacuum dried at 60 °C overnight (*p* < 70 mbar) and stored as CBB (yield: 335.0 mg, 73%). Elemental analysis for synthesized CBB in % (calc.) was as follows: Cs 21.5 (26.0); Bi 27.1 (27.2); Br 35.6 (46.8). 

### 2.4. Synthesis of Cs_3_Bi_2_Br_9_/HTiO-NS (CBHTNS) Heterostructures

The preparation of CBHTNS heterostructures is illustrated in Figure 1b. The weight percentage (wt%) of CBB was varied to form a set of different heterostructures of CBB/HTiO-NS. First, 10 wt% CBB precursor solution was prepared using 40.4 mg of CsBr and 56.7 mg of BiBr_3_ in 4.2 mL DMSO (Mixture A). Mixture A was stirred at 500 rpm until there were no remaining solids. Next, 500.0 mg HTiO-NS was mixed with 500 mL IPA at room temperature (Mixture B). Mixture B was centrifuged for 30 min and vigorously stirred at 500 rpm for another 5 min. Next, mixture A was quickly injected into mixture B under vigorous stirring at room temperature and stirred continuously for an additional 10 min. Finally, the obtained mixture was centrifuged at 3000 rpm for 3 min and 10,000 rpm for 5 min using IPA. The obtained yellow solids were then vacuum dried at 60 °C (*p* < 70 mbar) overnight and stored as CBHTNS-10. The same procedure was repeated to synthesize the CBHTNS-30 (30 wt%), CBHTNS-50 (50 wt%) heterostructures. Elemental analysis for synthesized CBHTNS-10 in % (calc.) was as follows: Ti 45.8 (46.5); Cs 2.4 (4.3); Bi 2.2 (4.5); Br 3.6 (7.8), CBHTNS-30: Ti 35.3 (34.9); Cs 6.9 (9.7); Bi 8.1 (10.2); Br 12.2 (17.5) and CBHTNS-50: Ti 26.0 (28.0); Cs 9.0 (13.0); Bi 13.1 (13.6); Br 18.2 (22.3). 

### 2.5. Characterization Techniques

The Elemental analysis for Ti, Bi and Na was performed using an ICP-OES device (Anton Paar, Graz, Austria/Perkin-Elmer, Waltham, MA, USA) while Cs was determined by atomic absorption spectroscopy (AAS, PerkinElmer AAS-AAnalyst 300 spectrometers). Br content was determined by potentiometric titration (Titrator Excellence T7, Mettler Toledo, Columbus, OH, USA). 

SEM micrographs were recorded using a Merlin VP compact device (Zeiss, Oberkochen, Germany); the EDX was measured using a Bruker Quantax device, Billerica, MA, USA. TEM images were generated by using a JEOL, JEM-ARM200F, Freising, Germany which operated at an acceleration voltage of 200 kV.

Crystalline phases were measured via powder X-ray diffraction (pXRD, Xpert Pro diffractometer, Panalytical, Almelo, The Netherlands; Xcelerator detector with automatic divergence slits and CuKα_1_Kα_2_ radiation, 40 kV, 40 mA; λ_1_ = 0.15406 nm; λ_2_ = 0.15443 nm). The obtained intensities were converted from automatic to fixed divergence slits (0.25°) for further analysis. Finally, the phase identification was conducted using the database of the International Center of Diffraction Data (ICDD). The Scherrer equation was used to calculate mean crystallite sizes of the catalyst materials. 

The surface and chemical components were measured using X-ray photoelectron spectroscopy (XPS) which was recorded on a photoelectron spectrometer (Multilab 2000, Thermo Fisher, Waltham, MA, USA) using Al Kα radiation as the excitation source. All XP spectra were referenced to the C1s line at 284.6 eV.

The functional groups of the prepared catalyst were investigated using Fourier transform infrared spectroscopy (FTIR, Nicolet 330, Thermo Fisher Scientific, Waltham, MA, USA) under ambient conditions using KBr as the background.

The absorption edges of the prepared catalysts were measured via UV–Vis spectroscopy in diffuse reflectance (DRS, Lambda 365, PerkinElmer, Waltham, MA, USA) in a range of 190 to 1100 nm. The band gap energy of the photocatalysts was estimated via standard Tauc plot and the Kubelka–Munk equation. 

Photoluminescence (PL) spectra of the catalysts were determined by a fluorescence spectrophotometer (Agilent Technologies Inc., Mulgrave, Australia) at room temperature with an excitation wavelength of 350 nm in the range of 370 to 680 nm. The specific surface areas of all samples were determined by nitrogen physisorption according to the Brunauer–Emmett–Teller (BET) method using BELSORP max II (Microtrac Retsch GmbH, Haan, Germany) and a NOVAtouch (Anton Paar Germany GmbH, Ostfildern-Scharnhausen, Germany) devices. Before each measurement, the samples were degassed at 200 °C for 5 h.

Photoelectrochemical experiments were performed using the Zennium electrochemical workstation equipped with a PP211 CIMPS system (Zahner, Kronach, Germany) with a typical three-electrode cell. A 0.1 M tetra-n-butyl hexafluorophosphate in dichloromethane served as the electrolyte to investigate the CBB and HTiO-NS. The reference electrodes were saturated Ag/AgCl (3 M, NaCl), while the counter electrodes were platinum wire. To create a thin coating and a functional electrode, 20 mg of photocatalyst was dispersed in a mixture of 100 µL Nafion solution (5 wt%) and 900 μL IPA. The dispersion was applied on an active area of 1.5 × 1.5 cm^2^ ITO glass after being dispersed in ethanol for 10 min using ultrasonication. A 430 nm LED lamp (400 mW/cm^2^) was used as a light source for all photoelectrochemical tests. Electrochemical impedance spectroscopy (EIS) was carried out through a potential static method in the 0.01 to 100 kHz frequency range. Mott–Schottky curves were plotted to analyze the flat-band potentials of the photocatalysts by using the impedance received by the selector chemical. Conduction band (CB) and valence band (VB) potentials of n-type semiconductors HTiO-NS and CBB were calculated by using Equation (1). The $E_{FB}$ is the Fermi level energy vs. Ag/AgCl and the $E_{CB}$ is the CB energy, ${E^{o}}_{\left(\frac{Ag}{AgCl}\right)}$ is the standard electrode potential which is +0.209 V for Ag/AgCl at 25 °C at pH = 7.
(1)$$E_{CB}\left(NHE\right)=E_{CB}+0.059pH+{E^{o}}_{\left(\frac{Ag}{AgCl}\right)}$$

### 2.6. Evaluation of Photocatalytic Benzyl Alcohol (BnOH) Oxidation

Photocatalytic BnOH oxidation reactions were performed in 5 mL glass vials being magnetically stirred at 500 rpm under a 5 W blue light-emitting diode (LED, light intensity: 25 mw cm^−2^, maximum wavelength: 467 nm). A distance of 10 cm between the glass vials and the fan was fixed to maintain reactions at room temperature. Initially, 20 mmol L^−1^ of BnOH was prepared using acetonitrile as the solvent. For each experiment, 10 mg of catalyst was dispersed in 2 mL of BnOH solution. The solution was then purged with oxygen for 10 min at a 10.8 mL/min flow rate and then stirred at 500 rpm for 30 min in the airflow of the fan in the dark. Afterwards, the samples were irradiated with the 5 W blue LEDs under 500 rpm stirring for a fixed time period (2 h, 4 h and 6 h). Reactants and products were analyzed by gas chromatography (GC, Agilent 19091X-133) equipped with a flame ionization detector (FID). A capillary column with 30 m length, 0.25 mm inner diameter and 0.25 µm film thickness was used. The column temperature was set at 60 °C for 5 min and increased to 240 °C (rate: 15 K min^−1^) with a final holding time of 5 min. The chromatograms were obtained by injecting 1 µL of the sample. The quantitative results of oxidation products were based on the internal standard method. The typical analytical procedure is as follows: (1) after each reaction, the samples were taken and filtered (0.22 mm) for further evaluation; (2) the samples were diluted before the GC analysis. In each analysis sample, 0.5 µL of the sample was diluted with 0.5 µL of internal standard in the GC vial; (3) the diluted solution was injected into the GC instrument to analyze the oxidation products according to the different retention times and response peak area. For the photocatalytic stability test, the photocatalysts were collected via centrifugation and then dried for 12 h at 60 °C. The conversion, selectivity and yield percentage in the results section were calculated by application of Equations (2)–(5):(2)$$Conversion(\%)=\frac{C_{BnOH\;initial}-C_{BnOH\;after\;reaction}}{C_{BnOH\;initial}}\times 100\%$$
(3)$$Selectivity(\%)=\frac{C_{product\;after\;reaction}}{C_{BnOH\;initial}-C_{BnOH\;after\;reaction}}\times 100\%$$
(4)$$Yield\left(\%\right)=\frac{C_{product\;after\;reaction}}{C_{BnOH\;initial}}\times 100\%$$
(5)$$Conversion\;rate=\frac{Total\;yield\;(mg)}{mass\left(g\right)\cdot Time(h)}$$

### 2.7. Reactive Oxygen Species Monitoring (Scavenger Tests)

The same photocatalytic BnOH oxidation conditions and procedures as described above were applied during the scavenger tests. The scavenger tests were carried out to identify the main reactive oxygen species (ROSs) responsible for BnOH conversion. Potassium iodide (KI), silver nitrate (AgNO_3_), *p*-benzoquinone (*p*-BQ) and tert-butyl alcohol (TBA) scavengers were used to quench photogenerated holes (h^+^) [32], photogenerated electrons (e^−^) [33], superoxide radicals (•O_2_^−^) [28,34] and hydroxyl radicals (•OH) [32], respectively. The scavenger substance was added to each reaction sample with a fixed concentration of 3.0 mM. The scavenger activity influenced the photocatalytic activity as reflected in the mechanism of the related reactive species.

## 3. Results and Discussion

### 3.1. Elemental Analysis

Table 1 shows the weight percentage (wt%) in CBB, HTiO-NS and CBHTNS heterostructures determined by ICP-OES, AAS and potentiometric titration (halide content). The wt% of pristine CBB is 21.5 wt%, 27.1 wt% and 35.6 wt% for Cs, Bi and Br elements, respectively. Additional data from XPS and EDX measurements show for all materials that the balance wt% is exclusively oxygen. There are no metal impurities detectable in pristine CBB. The wt% for pristine HTiO-NS is 51.5 wt% of Ti and 1.4 wt% impurity of Na due to the excessive NaOH solution used during the hydrothermal process to change the morphology of spherical UV100 into a nanosheet structure of titanate. The higher the weight percentage of CBB added onto the HTiO-NS, the higher its Cs, Bi and Br wt%, and the lower the Ti wt% detected by the ICP. The CBHTNS heterostructures show only minor amounts of Na impurities (<0.5 wt%).

Figure 1 shows photographs of all prepared CBHTNS heterostructures. These photographs demonstrate that the higher the weight percentage of CBB, the brighter the yellowish color of the catalyst. The bright yellow color of CBB is similar, as reported by Donguk Lee et al., for nanoparticles of Cs_3_Bi_2_Br_9_, while nanocrystals of Cs_3_BiBr_9_ appear colorless [35].

### 3.2. Morphology and Structure Characterization

Surface morphologies and structures of pristine HTiO-NS, CBB and CBHTNS heterostructures were investigated by SEM and TEM. As shown in Figure 2a, the nanosheet structure of HTiO-NS was observed. The high smoothness of the developed sheet-like hydrogen titanate suggests a higher structural stability and photocatalytic efficiency [36,37]. The sheet-like CBHTNS structure in Figure 2b is similar to pristine HTiO-NS (Figure 2a) even after the CBB was introduced. Figure 2c,f shows the element percentage and elemental mapping in CBHTNS-30, measured using EDX. The elemental mapping (Figure 2f) shows the rather homogeneous distribution of Ti, O, Cs, Bi and Br elements to form this CBHTNS-30 heterostructure. The good spread of elements implies the efficient interfacial photogenerated charge diffusion on light irradiation and enhances the photocatalytic activity [28,37]. The TEM images of the CBHTNS-30 (Figure 2d,e) clearly show two-dimensional layered structures and the existence of CBB (arrowed and circled), while the less intense dark is the nanosheet structure of HTiO-NS. In conclusion, the SEM-EDX and TEM results prove the formation of CBHTNS heterostructures and further confirm the homogeneous distribution of CBB on HTiO-NS. 

pXRD patterns (Figure 3a) show the diffraction reflexes of the HTiO-NS, CBB and CBHTNS heterostructures with varying CBB loadings. A set of diffraction reflexes with 2θ ranging from 10 to 60° were observed for all photocatalysts. For instance, the HTiO-NS exhibited distinctive hydrogen titanate nanosheet reflexes located at 2θ = 24.19° (110) and 48.43° (020), which are in agreement with the diffraction of hydrogen tri-titanate (H_2_Ti_3_O_7_, JCPDS No. 41-0192) [37,38,39,40]. The hydrogen tri-titanate composes of basic skeleton of edge sharing triple [TiO_6_] octahedron and interlayer H+ which can be easily exchanged by other cations, leading to the good adsorptive properties of HTNS [41,42,43]. The diffraction reflexes of pristine CBB match very well with the characteristic pattern of Cs_3_Bi_2_Br_9_ (JCPDS 01-070-0493) as well as an earlier study, indicating that it has a high crystallinity and cubic structure [31,44,45]. The reflexes are located at 2θ = 15.65° (011), 22.19° (110), 27.42° (003), 31.68° (022), 38.96° (122), 45.26° (220) and 51.09° (131). No impurity was observed in the crystalline phase of the as-prepared CBB indicating high phase purity. The XRD patterns of all CBHTNS heterostructures exhibit two-phase diffraction reflexes, and the intensity of the perovskite reflexes increases with loading, similar to those reported by a previous study [28]. The bonding composition and functional groups of the dried catalysts were further investigated by FTIR spectroscopy (Figure 3b,c). The FTIR spectrum of HTiO-NS shows a broad band at 3800–3000 cm^−1^ and 1627 cm^−1^, which are assigned to the stretching vibration of H–O (hydroxyl groups) and binding vibrations of H–O–H (bound water), respectively [40,46,47]. The relatively low intensities of the H–O–H bending and O–H stretching modes in the CBHTNS heterostructures in comparison to HTiO-NS suggest that the association of CBB in the heterostructures may result in the removal of some water molecules from the surface of HTiO-NS [47]. Figure 3c shows the HTiO-NS and CBHTNS heterostructures has bands at 906 cm^−1^ and 440 cm^−1^, which are assigned to the vibration of Ti–O nonbridging oxygen bonds and Ti–O vibration which related with [TiO_6_] octahedron, respectively [48,49,50]. The CBB and CBHTNS heterostructures show three bands centered at approximately 1122 cm^−1^, 1014 cm^−1^ and 658 cm^−1^ which might correspond to the vibration of bismuth [51], cesium [52] and bromide [53] with carbonyl. It is important to note that all CBHTNS heterostructures have the same bands as pristine HTiO-NS and CBB, further confirming the successful construction of CBHTNS heterostructures.

XPS measurements were carried out to investigate the elemental compositions and surface chemical states of all CBHTNS heterostructures as depicted in Figure 4a. As depicted in Figure 4a, a low peak of C 1s was observed with less than 1 wt% (Table 1) in CBHTNS heterostructures, which might be adsorbed on the samples from the carbon-based materials in the atmosphere or from the residual solvent during the synthesis. The Ti 2p spectra (Figure 4b) show the HTiO-NS has two peaks with binding energies of 458.6 eV and 464.3 eV. The 458.6 eV indicates the Ti 2p_3/2_ species, and 464.3 eV is ascribed as Ti 2p_1/2_ [8]. The CBHTNS heterostructure shows similar binding energies but a positive shift in relation to the pristine HTiO-NS at 459.3 eV and 465.0 eV. The binding energies of Cs 3d (Figure 4c) show that the binding energies of CBB are located at 724.6 eV, which is ascribed to Cs 3d_5/2_ and at 738.5 eV to Cs 3d_3/2_ species, respectively [54,55]. The CBHTNS heterostructure shows similar binding energies at 724.6 eV and 738.5 eV. The XPS Bi 4f spectra (Figure 4d) show the CBB has two peaks with binding energies at 159.3 eV, ascribed to Bi 4f_7/2_, and at 164.6 eV Bi, 4f_5/2_ [56,57]. The XPS spectrum of CBHTNS heterostructure shows slight peaks negatively shift to 158.8 and 164.1 eV, respectively. The binding energies of Br 3d (Figure 4e) show the CBB and CBHTNS has the same two peaks with binding energies at 68.6 eV and 69.6 eV, similar to those reported previously [57]. In these results, the CBHTNS heterostructure shows a positive shift in relation to the pristine HTiO-NS for Ti 2p spectra (Figure 4b) and a slight negative shift for Bi 4f spectra (Figure 4d), which confirm the possible charge transfer from HTiO-NS to CBB in the CBHTNS heterostructures [58]. In conclusion, the XPS confirmed the successful construction of CBHTNS heterostructures and proved the synergic interaction between CBB and HTiO-NS via electron interaction. 

Nitrogen adsorption–desorption isotherms of the investigated materials are depicted in Figure 5. The adsorption–desorption isotherm results of HTiO-NS and CBHTNS heterostructures exhibited a typical IV isotherm with a high adsorption capacity in the high relative pressure range of 0.7–1.9 P/P_0_, indicating the presence of mesopores (2–50 nm) in the material [2,40,59,60]. The pore size of HTiO-NS and CBHTNS heterostructures of 2.8–2.0 nm is in the range of small pore types [61], which were assigned to an internal space of curly nanosheets [40]. The surface area of pristine HTiO-NS was determined to be 416 m^2^/g, while the surface areas decreased dramatically with an increase in CBB loading (CBHTNS-10: 108 m^2^/g, CBHTNS-30: 69 m^2^/g and CBHTNS-50: 23 m^2^/g). The results are similar to those reported by Bresolin Bianca-Maria et al., indicating that perovskite formation creates an excessive aggregation on the surface of the pristine TiO_2_ [62]. Generally, it is thought that the surface area influences the photocatalytic performance if the photocatalyst is the same [63]. The large surface area of HTiO-NS benefits the increase in the surface area of CBHTNS heterostructures and enhances the adsorption of BnOH. The lower performance of CBHTNT-10 compared to CBHTNS-30 and CBHTNS-50, even though it has a higher surface area, is obviously related to the lower active side, CBB with only 10 wt%.

### 3.3. Evaluation of Photocatalytic Activity—BnOH Oxidation

The conversion and selectivity of BnOH oxidation towards BzH and BzA for pristine CBB, HTiO-NS and without the presence of any catalyst are shown in Figure 6a, while the results for CBHTNS heterostructures are shown in Figure 6b. From the results, the following important observations could be made. First, no benzaldehyde (BzH) was detected for pristine HTiO-NS and without a catalyst. Below 20% of conversion to benzyl benzoate or benzene might be due to the irradiation of blue light, as previously reported in the literature [64]. Second, there was a high selectivity of BzH for pristine CBB. However, the selectivity of BzH was decreased with increased reaction time. Third, benzoic acid (BzA) was the major oxidation product for all CBHTNS heterostructures (Figure 6b). Compared to pristine CBB (Figure 6a), the oxidation products of CBHTNS heterostructures were BzH and BzA. For instance, CBB 2h showed BzH exclusively as the reaction product after 2 h of irradiation (84% selectivity), while CBHTNS-30 showed BzA and BzH product formation (BzA: 75% and BzH: 21%, respectively). This observation was similar to that reported by Stefan et al., who reported that the selectivity of BzH was decreased with the formation of BzA [27]. BzH is easier to convert to BzA because BzH is an intermediate product for BnOH oxidation to BzA [65]. This observation demonstrates the synergistic effect of CBHTNS heterostructures compared to pristine CBB and HTiO-NS. Fourth, CBHTNS-30 performed at the optimal conditions compared to other CBHTNS heterostructures. Adding 50 wt% of CBB to HTiO-NS for only 1% of increased product selectivity is not worth it. Increasing reaction times slightly reduced the product selectivity for CBHTNS heterostructures. For instance, CBHTNS-30 showed 75% selectivity towards BzA after 2 h of irradiation, compared to 71% after 6 h. This observation might be due to the prone decomposition properties of the CBB catalyst [66]. CBHTNS-10 showed BzH exclusively as a reaction product with low selectivity, which might be due to the low loading of CBB (10 wt%). The enhanced photocatalytic oxidation of CBHTNS-30 and CBHTNS-50 originated from the beneficial ways that HTiO-NS increases the oxidative properties, possibly by charge separation in the heterojunction. Therefore, it can be concluded that the enhanced BnOH photooxidation performance confirms the successful formation of CBHTNS heterostructures, which was discussed previously via a detailed discussion on the morphology (Figure 2f) and structure characterization (Figure 3, Figure 4 and Figure 5) (see above). Figure 6c shows the oxidation of BnOH of CBHTNS-30 at 2 h, in two consecutive cycles. The recycling experiments show relatively poor recyclability, with a reduction of BnOH conversion from 98% to 44%. The decreased BnOH conversion for the second cycle is aligned with CsPbBr_3_/TiO_2_ [27]. The poor recyclability was ascribed due to the poisoning by adsorbates (BzH and BzA), which accumulated during the reaction in the absence of water [27,67]. The post-reaction characterization of the CsPbBr_3_/TiO_2_ via UV/Vis, XRD and STEM studies confirmed that the band gap remains unchanged at 2.34 eV after the photocatalytic reaction. The XRD patterns of the used sample show no changes compared to the as-prepared catalyst and the STEM micrographs recorded the small CsPbBr_3_ nanoparticles decorated on the TiO_2_ support. This result indicates a good stability of CsPbBr_3_ under the reaction conditions, which further confirms that the poor recyclability is related to TiO_2_ rather than to the halide [27]. Conversely, the Cs_3_Bi_2_Br_9_/TiO_2_ heterojunction showed excellent stability of BnOH oxidation with the negligible changes in conversion rate and selectivity for the five catalytic cycles [28].

Figure 7a shows the photocatalytic BnOH oxidation performance of the optimal catalyst in this study compared to recently reported perovskites as well as titania- and hydrogen titanate-based photocatalysts from the literature. The details of the reaction condition are tabulated in Appendix A. It is important to note that the reaction conditions are barely comparable, thus affecting conversion, selectivities and reaction product formation. For instance, Qimeng Sun et al. reported that Cs_3_Bi_2_Br_9_/TiO_2_ nanosphere photocatalysts showed excellent catalytic activity under mild reaction conditions with a 1.5 mmol g^−1^h^−1^ conversion rate [28]. The conversion rate of CBHTNS-30 in this study is 1.2 mmol g^−1^h^−1^ (BzH) and 3.7 mmol g^−1^h^−1^ (BzA) which might be due to the presence of a nanosheet-like structure of HTiO-NS or the application of O_2_ in our reactions. In their recent study, Stefan S. et al. reported that CsPbBr_3_/TiO_2_ (Evonik P25) showed 40% conversion and excellent selectivity towards BzH (>99%). They found that at 50% conversion, the BzH selectivity was reduced to <90% and BzH was further oxidized into BzA [27]. This observation is comparable to this study, where the selectivity was reduced from 84% BzH (CBB) to 21% BzH and 75% BzA (CBHTNS-30; Figure 6a,b). Furthermore, Jianbo Jin et al. reported that Cs_2_TeBr_6_ and Cs_3_Bi_2_Br_9_ photocatalysts converted BnOH to BzH under a 365 nm LED and showed further oxidation towards BzA after a 2 h reaction [68]. The conversion to BzA oxidation products was also reported by Mingming Du et al., with 91% BnOH conversion, 72.6% BzH and 18.5% BzA selectivity using Au-Pd/H_2_Ti_3_O_7_ nanowire photocatalysts [64]. In our system, noble metals are not needed, which is an advantage. In the other titania-supported photocatalyst, only BzH oxidation products were formed. For instance, Xiaolei Bao et al. reported that the TiO_2_/Ti_3_C_2_ photocatalyst exhibited a 97% conversion efficiency and 98% BzH selectivity [69]. Xiong-Fei Zhang et al. reported that the Au_1_Pt_1_/TiO_2_ photocatalyst converted the BnOH into BzH with a high selectivity of >98% and 65.3% conversion efficiency [70]. JamJam et al. reported that the 0.5 wt% Au-0.5 wt% Pd/TiO_2_ photocatalyst performs well in BnOH oxidation with a 19% conversion efficiency and 80.5% of BzH selectivity [65]. These recent results confirmed that the titania- and hydrogen titanate-based photocatalysts are suitable and stable photocatalysts to enhance the performance of photocatalytic BnOH oxidation. In conclusion, the same trend of BnOH photocatalytic oxidation performance was reported by other studies as depicted in Figure 7a. The longer reaction times might convert the BnOH to BzA and reduce the selectivity of BzH. In contrast, the suitable heterojunction of CBHTNS in this study successfully converted the BnOH to BzA as a major reaction product under mild reaction conditions after 2 h; consequently, the low reaction cost might be conducive for scaling up purposes. The influence of solvent polarity was studied, and it was reported that the lower polarity showed a better BnOH conversion efficiency than the higher polarity of the solvent [69]. The molecules of a solvent with higher polarity will compete with the BnOH molecules to be adsorbed on the surface of the catalysts; the higher polarity solvent adsorbed more easily on the catalyst and prevents the subsequent absorption of BnOH, resulting in a decrease in reactive sites and lower the conversion efficiency of BnOH oxidation. Figure 7b shows some recent literature on the degradation performance of halide perovskite and hydrogen titanate photocatalysts, which are significant and comparable with the CBHTNS sample in this study. The details of the reaction condition are tabulated in Appendix A. It is clearly shown that the halide perovskite performed a high efficiency of photodegradation with 93% and 80% photodegradation efficiency for pure Cs_3_Bi_2_I_9_ [71] and Cs_3_Bi_2_Br_9_ [45], respectively. The 99% and 98% photodegradation efficiency was shown by Cs_3_Bi_2_Br_9_/TiO_2_ [28] and C_3_N_4_/Cs_3_Bi_2_Br_9_ [29], respectively. This study reported that the construction of halide perovskite heterostructure had improved the band alignment and charge carrier migration of CBB, improving the photocatalytic performance. Different studies reported a significant degradation performance of HTNMs with various organic dyes [40,72]. It is noteworthy that the photocatalytic degradation efficiency decreased with the increase in initial dye concentration [45]. Therefore, the CBHTNS heterostructure has the potential to perform an enhanced photodegradation performance compared to pristine CBB and HTiO-NS in this study.

### 3.4. Mechanism of the Enhanced Photocatalytic Performance

Figure 8 shows the UV–vis DRS of as-synthesized HTiO-NS, CBB and CBHTNS heterostructures to investigate their optical properties. The absorption edges in Figure 8a of HTiO-NS and CBB are 368 nm and 490 nm, respectively, similar to those previously reported [8,57]. The absorption edges of CBHTNS-10, CBHTNS-30 and CBHTNS-50 are 402 nm, 475 nm and 482 nm, respectively. This indicates the additional presence of CBB in the CBHTNS heterostructures lead to the absorption of the visible-light region compared to pristine HTiO-NS. The Kubelka–Munk function was used to draw a Tauc plot and estimate the band gap energies of HTiO-NS, CBB and CBHTNS heterostructures for a comparison with previous works. We note here that the Tauc plot is often not physically meaningful, since it is based on various assumptions not fulfilled for combinations of semiconductors with more than one absorption process [73]. But it can qualitatively allow conclusions on light absorption when materials are compared with one another. The band gap energy of HTiO-NS and CBB (Figure 8b) is 3.2 eV and 2.5 eV, respectively. The lowest optical transition in the composites corresponds to a hypothetical band gap energy of 3.1 eV for CBHTNS-10, 2.6 eV for CBHTNS-30 and 2.5 eV for CBHTNS-50. These results demonstrate the successful visible light absorption in HTiO-NS and CBB composites, beneficial for constructing optimal CBHTNS heterostructures while improving charge transfer efficiencies [22,28]. Figure 8c depicts the PL spectra of as-synthesized HTiO-NS and CBHTNS heterostructures. The peaks for HTiO-NS are located at 360 nm, 426 nm and 461 nm (see insert graph), while in the CBHTNS heterostructures, they are shifted to 486 nm, 532 nm and 542 nm. The PL observation indicates that the CBHTNS-30 heterostructure has a more effective interfacial charge separation to suppress the charge recombination rate than CBHTNS-10 and CBHTNS-50. This might be because the 30 wt% of CBB (CBHTNS-30) has less aggregation on the surface of HTiO-NS and has a bigger surface area compared to the CBHTNS-50 (see Figure 5), which affect the charge transfer efficiency [62]. Figure 8d shows the EIS of HTiO-NS, CBB and CBHTNS heterostructures to further investigate the interfacial charge transfer properties. In Nyquist plots, the magnitude of the semi-circular curve indicates the extent of the charge transfer resistance of the electrode surface, where a larger arc radius indicates higher resistance [74,75]. The arc resistance (−Z″) of CBB was relatively lower than that of HTiO-NS, indicating that it possessed better charge transfer efficiency [76]. The lower resistance in CBB enhanced the effectiveness of the electron–hole pair separation in CBHTNS heterostructures [76,77], as shown in the PL results (Figure 8c). The Mott–Schottky plots (Figure 8e,f) use the data obtained from the capacitance–voltage measurements. The Mott–Schottky slope of HTiO-NS and CBB showed an n-type semiconductor with a potential of −0.67 V and −1.21 V, respectively. By using Equation (6), the conduction band (CB) and valence band (VB) energy of HTiO-NS were calculated as shown below:$$E_{FB}=-0.67\;V\;E_{CB}=-0.77\;V,\;E_{CB}\;is\;0.1\;V\;more\;negative\;than\;E_{FB}$$
(6)$$E_{CB}\left(NHE\right)=E_{CB}+0.059\;pH+{E^{o}}_{\left(\frac{Ag}{AgCl}\right)}\;{E^{o}}_{\left(\frac{Ag}{AgCl}\right)}=0.209\;V\;and\;pH=7$$
$$E_{CB}\left(NHE\right)=-0.77\;V+0.059\times 7+0.209\;V=-0.148\;V$$
$$E_{VB}\left(NHE\right)=E_{CB\left(NHE\right)}+E_{g};\;band\;gap\;energy\;is\;3.23\;eV$$
$$E_{VB}=-0.148+3.23=+3.082\;V$$

Next, the CB and VB energy of CBB were calculated using the Equation (6) as shown above:$$E_{FB}=-1.21\;V\;E_{CB}=-1.31\;V,\;E_{CB}\;is\;0.1\;V\;more\;negative\;than\;E_{FB}$$
$$E_{CB}\left(NHE\right)=-1.31\;V+0.059\times 7+0.209\;V=-0.688\;V$$
$$E_{VB}\left(NHE\right)=E_{CB\left(NHE\right)}+E_{g};\;band\;gap\;energy\;is\;2.55\;eV$$
$$E_{VB}=-0.688+2.55=+1.862\;V$$

From the calculation using Equation (6), the CB (NHE) of HTiO-NS and CBB are −0.148 V and −0.688 V, respectively. The VB (NHE) energies of HTiO-NS are +3.082 V and CBB is 1.862 V, respectively. The calculated band structures of HTiO-NS and CBB are illustrated in Figure 8g. The superoxide radicals (•O_2_^−^) can be formed via a reduction of oxygen (O_2_) and singlet oxygen (^1^O_2_) at −0.33 V and +2.53 V, respectively [28]. The higher CB position of CBB −0.688 V) is suitable for the reduction of O_2_ to •O_2_^−^ at −0.33 V. The VB position of HTiO-NS (+3.082 V) is suitable to oxidize the BnOH at +1.90 V [28,68,78]. The VB position of CBB (+1.862 V) is appropriate to oxidize the hydrogen peroxide (H_2_O_2_) to water (H_2_O) at +1.763 V. Therefore, this might be the reason for the decreased product selectivity with the increased treatment hours of CBB (Figure 6a); the excess H_2_O formed might decompose the CBB. In conclusion, the well-matched band structure of CBHTNS heterostructures could separate the photogenerated e^−^, h^+^ and actively produce the •O_2_^−^ and •OH reactive species to improve the photocatalytic BnOH oxidation process as the observation shows in Figure 6. 

The scavenger’s results were evaluated to investigate the possible contribution of reactive species in CBHTNS photooxidation of BnOH, as in Figure 9a, to further suggest the mechanism of this reaction as in Figure 9b. In general, the declined photocatalytic performance in the presence of scavengers demonstrates the importance of all reactive species: photogenerated holes (h^+^), photogenerated electrons (e^−^), superoxide radicals (•O_2_^−^) and hydroxyl radicals (•OH). For instance, the reactive species of h^+^, e^−^, •O_2_^−^ and •OH were assumed to be captured by the scavenging compounds of KI, AgNO_3_, *p*-BQ and TBA, respectively. From the results, the following important observations can be made. First, no benzoic acid (BzA) was detected from the CBHTNS samples which were added with scavengers. All reactive species have been proven to affect the CBHTNS photooxidation of BnOH activity. Secondly, the KI, AgNO_3_ and *p*-BQ which captured the h^+^, e^−^ and •O_2_^−^ respectively, show a significant effect. For instance, the selectivity of CBHTNS without scavengers was significantly reduced from 21% (BzH) and 75% (BzA) to 15% (BzH) for CBHTNS KI, 22% (BzH) for CBHTNS AgNO_3_ and 27% (BzH) for CBHTNS *p*-BQ. Next, the CBHTNS TBA demonstrate the medium effect with a reduction to 62% (BzH). This indicates that the captured •OH by the TBA slightly affects the performance. Therefore, the scavengers’ results in Figure 9a allow us to conclude that the h^+^, e^−^ and •O_2_^−^ play an essential role similar to previous works [28]. It is important to note that some of the involved charge carriers (•OH, h^+^ with sufficient oxidation potential to oxidize BnOH) can only be formed on HTiO-NS, not on CBB. Although band gap excitation of HTiO-NS is not possible with blue light, we assume that electrons and holes are formed under irradiation, possibly by an absorption due to defect sites. As is visible in Figure 8a, light absorption shows a tailing to below 400 nm, which is often an indication for the presence of defects. We do not know whether states above the valence band, or below the conduction band, are responsible for the slight visible light absorption, so for the sake of simplicity, we will assume in the following processes that band gap excitation will occur. Figure 9b shows the schematic illustration of BnOH photooxidation when CBB and HTiO-NS are in contact. The suggested e^−^ flow is aligned with the XPS results in Figure 4. The Ef alignment generated an interfacial electric field (IEF), with a direction from HTiO-NS to CBB and band bending. The energy band edge of HTiO-NS was bent downward, whereas CBB was continuously bent upward toward the interface. The IEF and band bending acted as barriers to prevent the e^−^ from transferring from CBB to HTiO-NS and the h+ from HTiO-NS to CBB [23]. The photocatalytic performance of BnOH oxidation is directly affected by oxygen activation and the cleavage process for O–H and C–H bonds [64]. Under blue light irradiation, e^−^ and h+ were excited in HTiO-NS (defect states) and CBB (Reaction 1; R1). The photogenerated e^−^ in the CB (or in trap states) of HTiO-NS flew out to the VB of CBB, thus enhancing the separation of charge carriers and maintaining the h^+^ in VB of HTiO-NS (or in trap states) for a high oxidation potential. The enhanced separation of charge carriers in CBHTNS heterostructures compared to pristine HTiO-NS was reflected in the PL results (Figure 8c). This resulted in the accumulation of e^−^ in the CB of CBB and h^+^ in the VB of HTiO-NS. The photogenerated e^−^ in the CB of CBB reacted with O_2_ to produce •O_2_^−^ (R2), while the accumulation of h^+^ in VB of HTiO-NS could accelerate the C–H bond cleavage of BnOH to produce BnOH radicals [33] (R3). Simultaneously, the produced •O_2_^−^ cleaves the O–H bond in BnOH radicals to generate the BzH [79] (R4). The excess O_2_ cleaves the BzH to further oxidize to produce BzA (R5) the same as reported by previous studies [64,80]. Based on the scavengers’ results (Figure 9a), the •OH play the least role in this reaction. The •OH was produced as the reactions (R6) and (R7). The produced •OH will then react with the BzH to produce BzA (R8). According to several reports, there is a possible mechanism to produce hydrogen (H_2_) (R9); however, the limited experimental set up in this experiment does not allow us to collect and measure the produce gas phase from this reaction. The photogenerated e^−^ and h^+^ transfer pathway (R1–R9) is similar as previously reported [81,82]. Electron spin resonance (ESR) spectroscopy measurement was carried out to scavenge the •O_2_^−^ by the comparable photocatalysts in the literature. It was reported that the CsPbBr_3_//TiO_2_ composite under visible light illumination shows a strong ESR signal compared to P25, which confirmed the formation of •O_2_^−^ radicals [27]. The ESR signals of superoxide radicals were also observed from the Cs_3_Bi_2_Br_9_/TiO_2_ heterojunction [28].
(R1)$$CBHTNS+hv\to e^{-}+h^{+}$$
(R2)$$e^{-}+O_{2}↔{•O}_{2}^{-}$$
(R3)$${C_{6}H_{5}CH}_{2}OH+h^{+}↔•C_{6}H_{5}CHOH+H^{+}$$
(R4)$$•C_{6}H_{5}CHOH+{•O}_{2}^{-}↔{C_{6}H_{5}CHO}^{-}+•HO_{2}$$
(R5)$${C_{6}H_{5}CHO}^{-}+O_{2}\to C_{6}H_{5}COOH$$
(R6)$${•O}_{2}^{-}+2H^{+}\to H_{2}O_{2}$$
(R7)$$H_{2}O_{2}+e^{-}\to •OH+{OH}^{-}$$
(R8)$$•OH+C_{6}H_{5}CHO\to C_{6}H_{5}COOH+H^{+}$$
(R9)$${2e}^{-}+2H^{+}\to H_{2}$$

## 4. Conclusions

In this study, the straightforward design of the Cs_3_Bi_2_Br_9_/HTiO-NS (CBHTNS) heterostructure was successfully performed via the modified anti-solvent reprecipitation method. A different weight per cent (wt%) of CBB was added to the prepared precursor of HTiO-NS. The increase in CBB definitely enhanced the active site and the formation of optimal heterostructure. The sample 30 wt% CBB/HTiO-NS (CBHTNS-30) had a band gap of 2.56 eV and it had the lowest charge rate of e^−^/h^+^ recombination owing to its optimal heterostructure. These results demonstrate the successful visible light absorption in HTiO-NS and CBB composites, which are beneficial for constructing optimal CBHTNS heterostructures while improving charge transfer efficiencies. The CBHTNS-30 showed the highest benzyl alcohol photooxidation performance with 98% BnOH conversion and 75% benzoic acid (BzA) selectivity. BzH was highly generated for pristine CBB. However, the selectivity was decreased with increased reaction time, which indicates the instability of CBB. The BzA was the primary oxidation product for all CBHTNS heterostructures, which indicated that it is a potent oxidation agent. The presence of HTiO-NS increased the oxidative properties, possibly by charge separation in the heterojunction, which facilitated the generation of hydroxyl radicals and increased the generation of superoxide radicals. Therefore, the enhanced BnOH photooxidation performance confirms the successful formation of CBHTNS heterostructures. The results of the current study offer a simple synthesis of CBHTNS heterostructures, which can be considered a promising candidate utilized for photocatalytic organic synthesis. Future research can enhance the selectivity of photooxidation performance by employing different HTNM morphologies, such as nanotubes and nanowires, which might enhance the contact and cooperation between CBB and HTNM-based photocatalysts. 

## Data Availability

Data are contained within the article.

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
