# Peer review of "Enhancing the Photocatalytic Activity of Halide Perovskite Cesium Bismuth Bromide/Hydrogen Titanate Heterostructures for Benzyl Alcohol Oxidation"

_nanomaterials, 2024, doi:10.3390/nano14090752_

Round 1

Reviewer 1 Report

Comments and Suggestions for Authors

It is interesting results. However, in order to better clarify the mechanism, the manuscript is recommended for publication after minor revisions.

1. The related references of catalytic Oxidation Benzyl alcohol (BnOH) by Halide Perovskite should be added,such as J. Mater. Chem. 2008, 18, 1146.Nanoscale Adv. 2020, 2, 274;J. Environ. Sci. 2021, 104, 399。

2. Figure 2 d is not good quality.

3. The crystal structures of Halide perovskite Cs3Bi2Br9 and HTiO should be given.

4.  HRTEM image of Cs3Bi2Br9/HTiO-NS (CBHTNS) heterostructures should be provided.

5. If possible, ESR of superoxide radicals (•O2 - ) was suggested to add.

Comments on the Quality of English Language

polished

Author Response

We thank the reviewer for the consideration of our manuscript and helpful remarks. We updated and enhanced our manuscript in regard to the reviewer's comments as depicted in detail below. We've uploaded the updated manuscript with "track changes"-mode turned on to highlight any alterations we've added to our manuscript.

  1. The related references of catalytic Oxidation Benzyl alcohol (BnOH) by Halide Perovskite should be added,such as J. Mater. Chem. 2008, 18, 1146.Nanoscale Adv. 2020, 2, 274; Environ. Sci. 2021, 104, 399。

Answer: The manuscript has been updated with additional references, including the three references as suggested by the reviewer.

  1. Figure 2 d is not good quality.

Answer: Figure 2d has been updated. The Figure has also renamed as Figure 2c for further improvement.

  1. The crystal structures of Halide perovskite Cs3Bi2Br9 and HTiO should be given.

Answer: The database (ISCD) powder patterns of halide perovskite Cs3Bi2Br9 and HTiO have been provided in Figure 3a and in the text. The HTiO-NS exhibited distinctive hydrogen titanate nanosheet reflexes located at 2θ = 24.19° (110) and 48.43° (020). The CBB showed the reflexes located at 2θ = 15.65° (011), 22.19° (110), 27.42° (003), 31.68° (022), 38.96° (122), 45.26° (220) and 51.09° (131).

  1. HRTEM image of Cs3Bi2Br9/HTiO-NS (CBHTNS) heterostructures should be provided.

Answer: Figures 2 d and e have been revised with notation of CBB for clarity.   

  1. If possible,ESR of superoxide radicals (•O2-) was suggested to add.

Answer: ESR investigations of superoxide radicals (•O2-) are planned for a follow-up manuscript soon to be submitted. In this case we have added additional information from comparable literature references to this manuscript. 

Reviewer 2 Report

Comments and Suggestions for Authors

This paper presents a straightforward method to synthesize bismuth halide perovskite heterojunctions Cs3Bi2Br9/HTiO-NS (CBB/HTiO-NS, CBHTNS). The authors applied the heterojunction to the photocatalytic oxidation of benzyl alcohol (BnOH). And the heterojunction was characterized. The photocatalytic principle of benzyl alcohol was also analyzed. This work will inspire the strategy of constructing perovskite-based photocatalysts by modifying catalyst structure for suitable applications in transformation reactions. There are still flaws in the whole article, so I suggest major revisions.

1. There are many grammatical errors throughout the paper. Please make language changes throughout the article.

2. Modify the abstract. The importance of the study needs to be highlighted in the abstract.

3. There are too many paragraphs in the introduction. The logic between each paragraph of the introduction is poor, please reorganize the introduction section. The last paragraph of the introduction needs to be emphatically revised to highlight the research objectives of this work.

4. Experimental conditions and error bars need to be added in Figures 6 and 8(a). Photo layout should be optimized.

5. What is the innovation of this study? The catalyst concentration of 5 g/L is too high for photocatalysis research. What are the advantages of your material compared to other materials? Tables of degradation efficiency of benzyl alcohol by other photocatalysts are suggested to be supplemented.

6.What’s the Recycling performance of the catalyst?

7. The conclusions should focus on the summary of the study, main findings, and possible implications.

Comments on the Quality of English Language

Extensive editing of English language required

Author Response

We thank the reviewer for the consideration of our manuscript and helpful remarks. We have updated and enhanced our manuscript in regard to the reviewer's comments as depicted in detail below. We've uploaded the updated manuscript with "track changes"-mode turned on to highlight any alterations we've added to our manuscript.

  1. There are many grammatical errors throughout the paper. Please make language changes throughout the article.

Answer: The manuscript has been updated with correct grammar/format and proofreading by co-authors. If there are still severe mistakes presents in the manuscript we encourage the reviewer to provide a detailed list of all instances.

  1. Modify the abstract. The importance of the study needs to be highlighted in the abstract.

Answer: We have rewritten and modified the abstract in regard to the reviewer's comment. The significance of this research has been added and highlighted in the abstract.

  1. There are too many paragraphs in the introduction. The logic between each paragraph of the introduction is poor, please reorganize the introduction section. The last paragraph of the introduction needs to be emphatically revised to highlight the research objectives of this work.

Answer: The introduction has been updated with more information on the shortcomings and objectives of the current investigation and study.

  1. Experimental conditions and error bars need to be added in Figures 6 and 8(a). Photo layout should be optimized.

Answer: Figures 6 and 8a have been updated as suggested.

  1. What is the innovation of this study? The catalyst concentration of 5 g/L is too high for photocatalysis research. What are the advantages of your material compared to other materials? Tables of degradation efficiency of benzyl alcohol by other photocatalysts are suggested to be supplemented.

Answer: The achievements of our study have been highlighted and updated, including the comparison of our results with published data. In contrast, the concentration of the catalyst can be considered low due to the low wt% of CBB (active site) on the catalyst. Therefore, the initial concentration was put at the given certain "high" level. The table of degradation efficiency of benzyl alcohol by other photocatalysts has been moved to the Supplementary Information.

  1. What’s the Recycling performance of the catalyst?

Answer: Information about the recycling performance have been added as depicted in Figure 6c.

  1. The conclusions should focus on the summary of the study, main findings, and possible implications.

Answer: Conclusions have been updated and focused to summarize the study, its main findings, and possible implications as suggested.

Reviewer 3 Report

Comments and Suggestions for Authors

MS No: 

nanomaterials-2939996-peer-review-v1

Title:

Enhancing the Photocatalytic Activity of Halide Perovskite Cesium Bismuth Bromide / Hydrogen Titanate Heterostructures for Benzyl Alcohol Oxidation

Authors:     

Huzaikha Awang, Abdo Hezam, Tim Peppel and Jennifer Strunk

The present paper deals with the synthesis and characterization of bismuth halide perovskite heterojunctions Cs3Bi2Br9/HTiO-NS (CBB/HTiO-NS, CBHTNS). Their photocatalytic efficiency is tested towards benzyl alcohol (BnOH) oxidation under UV light. In general, it includes a strong physicochemical and photoelectrochemical characterization, however the mechanistic study is poor. In my opinion, it can be accepted for publication in Nanomaterials only after major revision.

Below are some specific comments:

·       The authors state that they used a 5 W blue LEDs. I would advise them to carry out a chemical actinometry in order to find out the exact radiation entering their photoreactor.

·       Fig.2. The specific TEM images do not provide any information considering the morphology of the photocatalytic materials. Please replace them with other of higher magnification.  

·       The stability of the present materials should be tested under consecutive experimental runs.

·       In order to emphasize more on the stability of the as prepared photocatalysts XPS data of the used sample should be added in the revised manuscript.

·       A Figure presenting the efficiency of the present photocatalysts towards degradation of other micropollutants (such as pharmaceuticals or endocrine disruptors) would increase the scientific value of the present study.

·       The effect of pH should be added to the revised manuscript.

·       The authors should add data considering the efficiency of the present materials in real water matrices, such as bottled water and wastewater.

Author Response

We thank the reviewer for the consideration of our manuscript and helpful remarks. We have updated and enhanced our manuscript in regard to the reviewer's comments as depicted in detail below. We've uploaded the updated manuscript with "track changes"-mode turned on to highlight any alterations we've added to our manuscript.

  1. The authors state that they used a 5 W blue LEDs. I would advise them to carry out a chemical actinometry in order to find out the exact radiation entering their photoreactor.

Answer: The actinometry test has been performed as suggested and the information has been added to the manuscript text (LED, light intensity: 25 mw cm-2, maximum wavelength: 467 nm).   

2. The specific TEM images do not provide any information considering the morphology of the photocatalytic materials. Please replace them with other of higher magnification.

Answer: Figure 2 (TEM) has been updated with the notation of CBB structure. 

  1. The stability of the present materials should be tested under consecutive experimental runs.

Answer: Information about the recycling performance have been added as depicted in Figure 6c.

  1. In order to emphasize more on the stability of the as prepared photocatalysts XPS data of the used sample should be added in the revised manuscript.

Answer: Thank you specifically for this important suggestion. We will consider this in our follow-up manuscript which will be soon submitted. Herein, we have added more information in this regard as already  reported in the literature about the post-reaction characterization. 

The post-reaction characterization of the CsPbBr3/TiO2 via UV/Vis, XRD and STEM studies confirmed that the band gap remains unchanged at 2.34 eV after the photocatalytic reaction. The XRD patterns of used sample shows no changes compared to the as-prepared catalyst and the STEM micrographs recorded the small CsPbBr3 nanoparticles decorated on the TiO2 support [29].

  1. A Figure presenting the efficiency of the present photocatalysts towards degradation of other micropollutants (such as pharmaceuticals or endocrine disruptors) would increase the scientific value of the present study.

Answer: Thank you specifically for this important suggestion. We will consider this in our follow-up manuscript which will be soon submitted, too. We have added more information from the literature in regard to the photodegradation performance by comparable photocatalysts as depicted in table S2 and Figure 7b, respectively.

  1. The effect of pH should be added to the revised manuscript.

Answer: The influence of pH on the BnOH photooxidation has been rarely discussed in the literature. We have added more information in this regard from the literature to our manuscript. The influence of solvent polarity has been studied, and it has been reported that the lower polarity has shown a better BnOH conversion efficiency than the higher polarity of the solvent [86].

  1. The authors should add data considering the efficiency of the present materials in real water matrices, such as bottled water and wastewater.

Answer: We have not performed any specific experiments in "real" water environments due to the rather instability of CBB against multiple factors including specifically water. Our study was focusing on obtaining general information about the performance of the composite in water-containing environments. As soon as a composite will be obtained which shows complete stability towards water it will open new research in bottled or waste water environments, too.

Round 2

Reviewer 2 Report

Comments and Suggestions for Authors

Based on their earnest attitudes, the authors provided reasonable responses to most of the questions raised before, and the whole paper has been carefully reviewed and modified. I suggest acceptance.

Reviewer 3 Report

Comments and Suggestions for Authors

Accept in present forn